# Genetic Variation in *Schizothorax kozlovi* Nikolsky in the Upper Reaches of the Chinese Yangtze River Based on Genotyping for Simplified Genome Sequencing

**DOI:** 10.3390/ani12172181

**Published:** 2022-08-25

**Authors:** Jiayang He, Zhi He, Deying Yang, Zhijun Ma, Hongjun Chen, Qian Zhang, Faqiang Deng, Lijuan Ye, Yong Pu, Mingwang Zhang, Song Yang, Shiyong Yang, Taiming Yan

**Affiliations:** College of Animal Science and Technology, Sichuan Agricultural University, Chengdu 611130, China

**Keywords:** *Schizothorax kozlovi* Nikolsky, genetic diversity, population structure, Yangtze River, simplified genotyping by sequencing (GBS)

## Abstract

**Simple Summary:**

*Schizothorax kozlovi* Nikolsky is a unique cold−water fish in the upper reaches of the Yangtze River in China and has high economic value. In our study, genetic diversity and population structure analyses were performed on seven wild populations in the upper reaches of the Yangtze River by GBS. The above results indicate that the populations of *S. kozlovi* have different degrees of tolerance and selection pressure in response to temperature and altitude. The Wujiang population was genetically differentiated from the Jinsha River and Yalong River populations. The Wujiang intrapopulation has greater genetic diversity and differentiation than the Jinsha River and Yalong River populations, which demonstrates that the Jinsha and Yalong populations require more attention and resources for their protection. The results of this study will increase our understanding of the diversity of *S. kozlovi* in the upper reaches of the Yangtze River and provide a basis for the conservation and utilization of wild resources.

**Abstract:**

*Schizothorax kozlovi* Nikolsky is a unique cold−water fish in the upper reaches of the Yangtze River in China and has high economic value. In our study, genetic diversity and population structure analyses were performed on seven wild populations (originating from the Jinsha River, Yalong River, and Wujiang River) in the upper reaches of the Yangtze River by genotyping by sequencing (GBS). The results indicated that a total of 303,970 single−nucleotide polymorphisms (SNPs) were identified from the seven wild populations. Lower genetic diversity was exhibited among the intrapopulations of the three tributaries, and the Wujiang River population had significant genetic differentiation when compared to the Jinsha River and Yalong River populations. Furthermore, the selected SNPs were enriched in cellular processes, environmental adaptation, signal transduction, and related metabolic processes between the Wujiang population and the other two populations. The above results indicate that the populations of *S. kozlovi* have different degrees of tolerance and selection pressure in response to temperature and altitude. The Wujiang intrapopulation has greater genetic diversity and differentiation than the Jinsha River and Yalong River populations, which demonstrates that the Jinsha and Yalong populations require more attention and resources for their protection. The results of this study will increase our understanding of the diversity of *S. kozlovi* in the upper reaches of the Yangtze River and provide a basis for the conservation and utilization of wild resources.

## 1. Introduction

*Schizothorax kozlovi* (original Schizothoracinae fish) is a cold−water fish that has adapted to the subalpine environment at the edge of the Tibetan Plateau over a long period of time. Additionally, *S. kozlovi* is a unique fish and is mainly distributed in the upper reaches of the Yangtze River in China, including the Jinsha River, Yalong River, Chishui River, and Wujiang River [1]. Multiple factors have led to the sharp decline in *S. kozlovi* population resources and the stability and sustainability of population development, such as climate change, overfishing, water pollution, water conservancy, and hydropower project construction [2]. Therefore, analysis of the genetic characteristics of the *S. kozlovi* wild populations is helpful in monitoring and improving the development level of the aquatic ecological environment in the upper reaches of the Yangtze River. Previous studies of the growth and breeding characteristics [3], diets and resource protection [4], and genetic diversity [5] of *S. kozlovi* have focused on the upstream Wujiang River. To date, the genetic characteristics of *S. kozlovi* in the Jinsha and Yalong Rivers have not been reported.

At present, mitochondrial genes are mostly used to study the genetic diversity of Schizothorax species [6,7]. The cytochrome b (cytb) gene has better resolution in the discrimination of Schizothoracinae fish than the cytochrome c oxidase subunit I (COI) gene [7]. However, there are still some confusing clustering relationships, which might have greatly weakened the resolution in the discrimination of Schizothorax species based on the cytb gene [7]. Furthermore, the cytochrome b (1141 bp) gene and the control region (712 bp) were used to analyze the phylogeography of *S. waltoni* in the Yarlung Tsangpo River (YLTR) on the southern margin of the Qinghai–Tibet Plateau [8]. However, there are some limitations to the use of mitochondrial genes to reveal population evolution, such as the relatively small size of the mitochondrial genome in the whole fish genome and nuclear mitochondrial pseudogene sequence contamination or sequencing errors [7,9].

Genotyping by sequencing (GBS), as one of the simplified genome sequencing approaches, can identify many single−nucleotide polymorphisms (SNPs) to reveal the evolutionary relationship between species and population genetics [10,11,12]. Unlike single−molecule DNA sequencing methods, a highly promising GBS approach can discover tens of thousands of variants and obtain genotypic information at the genomic level [13]. GBS reduces the complexity of genome sequencing through restriction endonucleases that are particularly suitable for species with large genomes, high repetition frequencies, and limited genomic resources [13]. Furthermore, SNPs from GBS are much more abundant, have lower mutation rates, and can be genotyped with lower error rates [14]. In previous studies, GBS has been used to solve a series of evolutionary biological problems, such as fish genetic protection and local adaptation of different populations [15,16].

The present study analyzed the genetic diversity and population structure of seven wild populations of *S. kozlovi* originating from three tributaries, including the Jinsha River (3 populations), Yalong River (3 populations), and Wujiang River (1 population), in the upper reaches of the Yangtze River by GBS. The results could provide important evidence for germplasm resource protection and the development and utilization of *S. kozlovi*.

## 2. Materials and Methods

### 2.1. Sample Collection

A total of 51 samples of *S. kozlovi* were randomly selected from seven wild populations at the three tributaries of the upper reaches of the Yangtze River in China (from 2017 to 2019), including the Jinsha River (Luoxu (LX), Zhubalong (ZBL), and Shigu (SG)), Yalong River (Xinlong (XL), Yajiang (YJ), and Muli (ML)), and Wujiang River (Zongjihe (ZJH)) (Figure 1 and Table 1). Otherwise, *Schizopygopsis malacanthus* and *Gymnodiptychus pachycheilus* were collected from the Yalong River and sequenced as outgroups by GBS to construct phylogenetic trees. *S. kozlovi*, *S. malacanthus,* and *G. pachycheilus* species were identified based on a reference book [17]. Each sample included composite samples of a pair of fins on one side and back muscles and was stored in 75% alcohol and a −20 °C refrigerator until use. Genomic DNA was extracted by the traditional phenol–chloroform extraction method.

### 2.2. Library Preparation

Genomic DNA (gDNA) was extracted from frozen tissue (TIANcombi DNA Lyse & Det PCR Kit, Tiangen Biotech, Beijing, China). The quality of gDNA was evaluated by 1% agarose gel electrophoresis, and the concentration was tested using a NanoDrop 2000 spectrophotometer (Thermo Scientific, Boston, MA, USA). The optimal concentration (50 ng/μL) of all gDNA samples was utilized to construct the sequencing library by superGBS technology [18].

### 2.3. Sequencing and Genotyping

All gDNA samples were sequenced by the GBS method [18], which was performed by the OE Biotech Co., Ltd. (Shanghai, China). Briefly, gDNA samples were digested by the restriction enzymes *pst*I−HF and *msp*I (NEB, Beverly, MA, USA). The digested gDNA fragments were linked by T4 ligase plus to connect barcode adapters at both ends of the fragment. Then, an improved magnetic bead recovery system was used to recover 300–700 bp fragments. The recovered fragments were amplified by PCR using high−fidelity enzymes. Furthermore, Qubit was used to determine the concentration of PCR products, and only concentrations greater than 5 ng/μL were used. Finally, PCR amplification fragments were sequenced on an Illumina Nova, PE150 (Illumina, Inc., San Diego, CA, USA). The raw reads were obtained after removing the potential adaptor sequences by using STACKS [19] software. The clean reads were obtained by FASTP [20] software after filtering the redundant reads. The genome sequence of *Oxygymnocypris stewartia* [21] was used as the reference genome. Then, the clean reads were mapped to the reference genome to analyze SNP markers by BWA [22].

### 2.4. Genetic Diversity and Population Structure Analysis

The parameters of genetic diversity were analyzed by VCFTOOLS [23], including the expected heterozygosity (*H*_E_), polymorphism information content (*PIC*), observed heterozygosity (*H*_O_), and nucleotide diversity (*π*). ARLEQUIN 3.5.1.3 [24] was used to calculate F−statistics (*F*_st_). Population structure was determined according to the K value corresponding to the minimum value of the CV error (cross−validation error) using ADMIXTURE v1.3.0 (Alexander, CA, USA) [25], and sometimes the K value was set from 2–7. Population gene exchange and differentiation analyses for the three tributaries (Jinsha River, Yalong River, and Wujiang River) were performed by Treemix V1.12 (Pickrell, Chicago, IL, USA) [26].

### 2.5. Reconstructing the Phylogenetic Tree

The neighbor−joining (NJ) method [27] was utilized to construct the phylogenetic tree and differentiate populations. *S. malacanthus* and *G. pachycheilus* were selected as outgroups to construct the phylogenetic trees. The distance matrix was calculated by TreeBest [28] software, and the reliability of the NJ tree was tested by the bootstrap method (repeated 1000 times) [29]. Then, the FigTree program (http://tree.bio.ed.ac.uk/software/figtree/, accessed on 6 May 2021) was used to visualize the phylogenetic tree. Plink2 [30] software was used for the principal component analysis (PCA) of the SNP markers obtained, and the two feature vectors with the greatest influence were obtained.

### 2.6. Functional Enrichment of the Selected SNPs

The selected SNPs were screened by Bayescan v2.1 (Foll, Grenoble, France) [31] software. The parameter was set to PR_odds 10 (the prior probability of the neutral model was set to 10), which was consistent with the default software parameter. SNPs with *p*-values less than 0.05 after FDR (Benjamini-Hochberg) correction in the resulting data were identified as selected sites. For functional enrichment analysis, all selected SNPs were mapped to terms in the GO and KEGG databases. Then, significantly enriched GO terms and KEGG pathways were searched for among the selected SNPs using *p* < 0.05 as the threshold. GO terms were cataloged into three subgroups, including biological processes (BPs), cellular components (CCs), and molecular functions (MFs).

## 3. Results

### 3.1. GBS Data and SNP Discovery

From the Illumina HiSeq sequencing platform, 41.01 Gb raw reads were obtained from 51 samples (The original sequencing data are provided in the Appendix A). Then, after removing the tags with insufficient sequencing depth, a total of 39.0 Gb of clean reads were retained and used for subsequent analysis. The Q20 and Q30 values were ≥95.93% and ≥89.66, respectively. The average GC content was approximately 45%. A 94.04% similarity was observed when the simplified genome sequences of *S. kozlovi* were compared to the reference genome sequences of *O. stewartia*.

### 3.2. Clustering Analysis of Differential SNPs

A total of 303,970 SNPs were obtained by mutation detection and filtering (Dataset S1 and Dataset S2). The results show that the types of DNA base mutations included transition and transversion. Transition (AT→GC or GC→AT) was the main mutation type (Figure 2A). Furthermore, most SNPs were located in intergenic and intron regions and were next derived from the exon, downstream, and upstream regions (Figure 2B).

### 3.3. Analysis of Population Genetic Diversity

According to SNP data, the values of *H*_E_, *H*_O_, *π*, and *PIC* were 0.04901–0.07821, 0.09175–0.10054, 0.07452–0.09202, and 0.03676–0.06128, respectively (Table 2). The average values of *H*_E_, *H*_O_, *π*, and *PIC* of the Wujiang population were greater than those of the Jinsha and Yalong populations. The results showed that the genetic diversity of the Wujiang population was higher than that of the Jinsha and Yalong populations.

The *F*_st_ value and genetic distance among populations were 0.0038–0.1890 and 0.0038–0.2095, respectively. The *F*_st_ value and the genetic distance values were 0.1791–0.1824 (*p* < 0.05) and 0.1974–0.2073 between the Wujiang (ZJH) and Jinsha populations, respectively. Furthermore, the *F*_st_ values and the genetic distance values were 0.1580–0.1890 and 0.1720–0.2095 between the Wujiang (ZJH) and Yalong populations, respectively. Thus, there was a large genetic differentiation between Wujiang (ZJH) and the other two populations, with the highest *F*_st_ value (0.15 < *F*_st_ < 0.25) and genetic distance (Table 3). The ML population of the Yalong River had moderate genetic differentiation from the three Jinsha populations. The genetic diversity within the Jinsha and Yalong populations was low. However, the genetic distance within Jinsha populations was the smallest.

### 3.4. Phylogenetic Analysis

The NJ phylogenetic tree was constructed based on 303,970 SNPs (Figure 3). According to the outgroups *S. malacanthus* (SC69) and *G. pachycheilus* (SC70), the Jinsha and Yalong populations could not be separated. Specifically, Wujiang (ZJH) and the other two populations were clustered into different branches. The results show that Wujiang (ZJH) and the other two populations had significant differentiation.

### 3.5. PCA

According to the 303,970 SNPs, the PCA results showed that the contribution rates of the first principal component (PC1) and the second principal component (PC3) were 47.15% and 21.05%, respectively (Figure 4A). The Jinsha and Yalong populations were clustered into one group, and the Wujiang (ZJH) population was clustered separately. When the K value was 2 based on the CV error, the Wujiang (ZJH) population and two other populations (Jinsha River and Yalong River) could be identified (Figure 4B). Furthermore, the Wujiang (ZJH) population was the first independent population, and gene flow existed between the Jinsha River and Yalong River populations throughout history (Figure 4C). The above results suggest that the Wujiang (ZJH) population of *S. kolzovi* had genetic differentiation from the Jinsha and Yalong populations.

### 3.6. Functional Annotations of the Selected SNPs

The functional annotations of the selected SNPs were analyzed based on KEGG signaling pathways and GO terms. The differential KEGG pathways (Dataset S3) and GO terms (Dataset S4) of the selected SNPs between the two groups were screened by *p*-values ≤ 0.05. A total of 30, 18, and 23 significant KEGG pathways were identified in the YL vs. WJ, JS vs. WJ, and JS vs. YL groups, respectively (Figure 5A). Then, eight common pathways were found in the YL vs. WJ, JS vs. WJ, and JS vs. YL groups, including cellular process (apoptosis−fly (4 genes), cell cycle (11 genes), and p53 signaling pathway (6 genes)), environmental adaptation (circadian rhythm (2 genes)), and signal transduction (MAPK signaling pathway (22 genes), Notch signaling pathway (8 genes), Hippo signaling pathway (7 genes), and TGF−beta signaling pathway (7 genes)) (Figure 5B). The Hippo signaling pathway (7 genes) and Notch signaling pathway (4 genes) were found in the JSJ vs. WJ and JS vs. YL groups. Specifically, the Notch signaling pathway was common in the YL vs. WJ, JS vs. WJ, and JS vs. YL groups (Figure 5B).

There were 134 significant GO terms in the JS vs. WJ groups, and 23 common terms were identified in the YL vs. WJ and JS vs. WJ groups (Figure 5C). Then, the 23 common terms were classified into molecular functions (6 terms), cellular components (3 terms), and biological processes (14 terms) (Figure 5D). Several important GO terms were screened, such as mitochondrial translation, mitochondrial transport, regulation of translation, RNA splicing, peptidyl−serine phosphorylation, thyroid hormone receptor binding, and transmembrane transporter activity.

## 4. Discussion

To date, GBS technology has not been widely utilized for the analysis of Schizothoracinae fish. In our study, 41.01 Gb of raw data and 39.0 Gb of clean data were obtained from 51 samples on average. The average value of each sample was 0.76 Gb, which was larger than the 0.63 Gb of *Cynoglossus semilaevis* [32]. The sequencing data were of high quality, with an average mapping rate of 94.02%, Q20 ≥ 95.93%, and Q30 ≥ 89.66%. The Q20 and Q30 values of GBS data from *S. kozlovi* were greater than those of GBS data from *C. semilaevis* (Q20 ≥ 93.74% and Q30 ≥ 87.01%) [32]. Furthermore, the 303.970 SNPs from the present study were far more abundant than those from the mitochondrial control region [33] and the amplified fragment length polymorphism (AFLP) marker [2] of *S. kozlovi*. These data objectively reflect the advantages of high efficiency and high recovery of GBS technology. Otherwise, SNPs obtained from GBS technology were characterized by high density and uniformity in distribution on the genome [34]. Therefore, compared with traditional analytical methods, the results of this study can more comprehensively and accurately reveal the genetic characteristics of the species of interest and provide a reference for the development of related research in the future.

As an important indicator of evolutionary potential, a higher level of genetic diversity signifies stronger adaptability to environmental change [35,36]. Polymorphic information content (*PIC*) can reflect the degree of population genetic variation, such as a highly polymorphic locus with *PIC* > 0.5, a weakly polymorphic locus with *PIC* < 0.25, and a moderately polymorphic locus with 0.25 < *PIC* < 0.5 [37]. Then, the levels of average heterozygosity and genetic diversity were positively correlated with the capacity to adapt to the environment [38]. The average values of observed heterozygosity (*H*_O_ 0.09578) and expected heterozygosity (*H*_E_ 0.06743) of the three river populations were lower than those of *Schizothorax wangchiachii* (*H*_O_ 0.2007 and *H*_E_ 0.3160) and *Schizothorax lissolabiatus* (*H*_O_ 0.2695 and *H*_E_ 0.2892) [39]. The above data show that the Yalong River, Jinsha River, and Wujiang River intrapopulations showed lower polymorphism and genetic diversity. The reason might be related to the decrease in the wild population of *S. kozlovi* due to changes in the aquatic environment and human fishing. The populations in the upper reaches of the Jinsha River and the middle reaches of the Yalong River are overfished in terms of their growth and recruitment. The population resources of *S. kozlovi* in the upper reaches of the Wujiang River are also in a state of overexploitation [2,5], resulting in a decrease in the number of wild *S. kozlovi* year by year. In addition, the number of parents of *S. kozlovi* decreased following a decrease in the number of alleles in the offspring population and the occurrence of a low degree of genetic variation [3]. These factors may result in low heterozygosity, serious inbreeding, and decreased genetic diversity of *S. kozlovi*.

The level of genetic differentiation can be determined by *F*_st_ values [40]. According to the classification criteria of genetic differentiation−based *F*_st_ values, Wujiang vs. Yalong and Wujiang vs. Jinsha had relatively high genetic differentiation (0.15 < *F*_st_ < 0.25), which indicates that genetic differentiation of the Wujiang population had occurred. In addition to the Jinsha River in the upper reaches of the Yangtze River, *S. kozlovi* is also distributed in the upper reaches of some water systems of the Yunnan–Guizhou Plateau. The geographical distance between the Yunnan–Guizhou Plateau and the population of the upper Yangtze River is large, which indicates that the geographical distance has a great influence on the degree of genetic differentiation of the natural populations of *S. kozlovi*. Geographic proximity suggests more gene exchange and a lower degree of genetic differentiation. Similar results have also been verified in *Schizothorax prenanti* [41], *Schizothorax biddulphi* [42], and *Schizothorax molesworthi* [43]. Furthermore, the gene exchange of *S. kozlovi* distributed in the upper reaches of the Yangtze River was hindered to some extent by hydropower development. Thus, the Wujiang population had large genetic differentiation and low gene exchange with the Jiansha and Yalong River populations. The above result suggests that scientific management measures should be taken to increase the number of wild populations and the genetic diversity of *S. kozlovi*.

Temperature is a key environmental factor affecting species distribution. The temperature selection strategies of *Salvelinus fontinalis* in different water layers reduced the overlap of the same ecological niche through intraspecific temporal and spatial isolation, which led to the maximization of the use of feed resources in the environment and enhanced the overall ability of the population to adapt to the environment [44]. This difference in intraspecific temperature selection behavior was also found in other fish species [44,45,46]. For example, the small tributaries were rich in bait and had suitable temperatures, with growth faster for *Oncorhynchus mykiss* [47]. However, *O. mykiss* could form a competitive monopoly in the heat−sheltered area with lower−than−average water temperature through cluster behavior in the main stream of a river [47]. The above behavior of *O. mykiss* enhanced the overall environmental adaptability of the population and maximized the population’s growth rate [47]. The temperature and altitude of different habitats reflect the differences in their adaptation to the environment [48]. Previous studies have shown that the genetic isolation of *Diptychus maculates* is significantly related to temperature, and the large variation in environmental factors leads to weak gene flow in the species [48]. There was information that showed that the average temperature of the low−altitude upper Wujiang River in the past ten years (approximately 15 °C) was significantly higher than that of the high−altitude upper Jinsha River (approximately 2.5 °C) and middle Yalong River (0.64 °C) [49,50,51]. Therefore, these results suggest that temperature may be one of the reasons for the genetic differentiation of *S. kozlovi* populations between the low−altitude upper Wujiang River and the high−altitude upper Jinsha River and Yalong River.

Loci with high genetic differentiation values were interpreted as having undergone natural selection, which could reflect the selection pressure from certain factors. To date, several molecular pathways have been correlated with the high−altitude adaptation of fish. Several types of functional selection genes are involved in the regulation of temperature changes by processes such as signal transduction, energy metabolism, and membrane [44]. Specifically, transcriptome research on *Gymnocypris namensis* [52] and *Gymnocypris selincuoensis* [53] demonstrated similar results. In our study, the common molecular pathways of selected SNPs among the YL vs. WJ, JS vs. WJ, and JS vs. YL groups of *S. kozlovi* populations were enriched in the cellular process (3 signaling pathways), environmental adaptation (circadian rhythm), signal transduction (4 signaling pathways), and related metabolism GO terms (such as mitochondrial translation and mitochondrial transport). The temperature−related SNPS found in this study are mainly involved in the functions of genes, including mitochondrial translation and transport, energy metabolism, etc. It is possible that temperature is a stimulus to *S. kozlovi*, causing various physiological reactions, and thus gene transcription has become very active. High mitochondrial gene concentration has been studied in other cold−adapted ectotherms. Windisch demonstrated that the mitochondrial ribosomal protein in *Pachycara brachycephalum* has a high transcription rate under cold conditions, which has a positive effect on the adaptation of Antarctic fish to cold environments [54]. In addition, increased energy requirements under temperature stress are usually required to maintain normal body function, thus allocating less energy to growth, storage, and reproduction [55]. In aquatic ectotherms such as mussels, extremely high temperatures will lead to a reduction in aerobic range or induce a reduced metabolic rate [56]. Thus, these results signify that adaptive differentiation−related regions have been generated in the genome of *S. kozlovi* populations and are driven by differences in environmental factors such as temperature, leading to differences in physiological regulation among different populations.

## 5. Conclusions

In summary, our study analyzed the genetic diversity and population differentiation in three different populations of *S. kozlovi*, including those from the Jinsha River, Yalong River, and Wujiang River, based on GBS technology. Based on the analysis of selected SNPs, low genetic diversity existed in each population. However, there was higher genetic differentiation between the Wujiang River population and the other two populations (Jinsha River and Yalong River). Furthermore, the most selected SNP markers were enriched in cellular processes, environmental adaptation, signal transduction, and related metabolic processes. Although the Xinlong population had only 1 sample, it did not affect the overall analysis results of the Yalong River population. Our research helps to reveal the diversity level of *S. kozlovi* in the upper reaches of the Yangtze River and provides the molecular basis for the formulation of biological protection measures.

## Figures and Tables

**Figure 1 animals-12-02181-f001:**
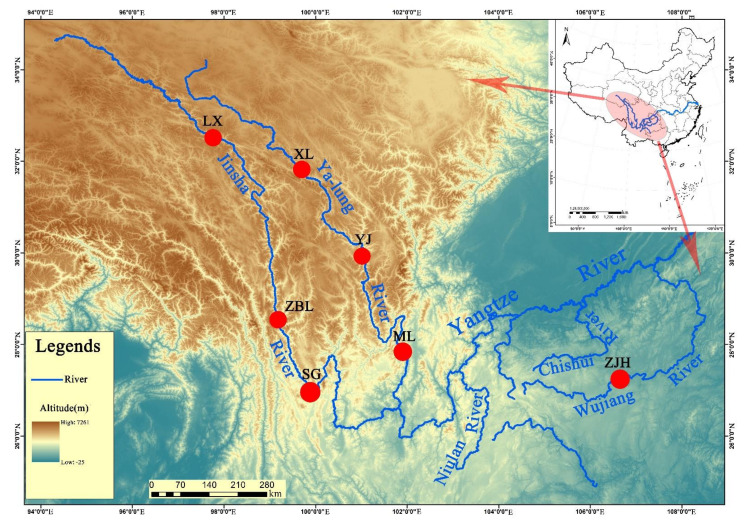
Map of sampling sites. LX, Luoxu; ZBL, Zhubalong; SG, Shigu; XL, Xinlong; YJ, Yajiang; ML, Muli; ZJH, Zongjihe.

**Figure 2 animals-12-02181-f002:**
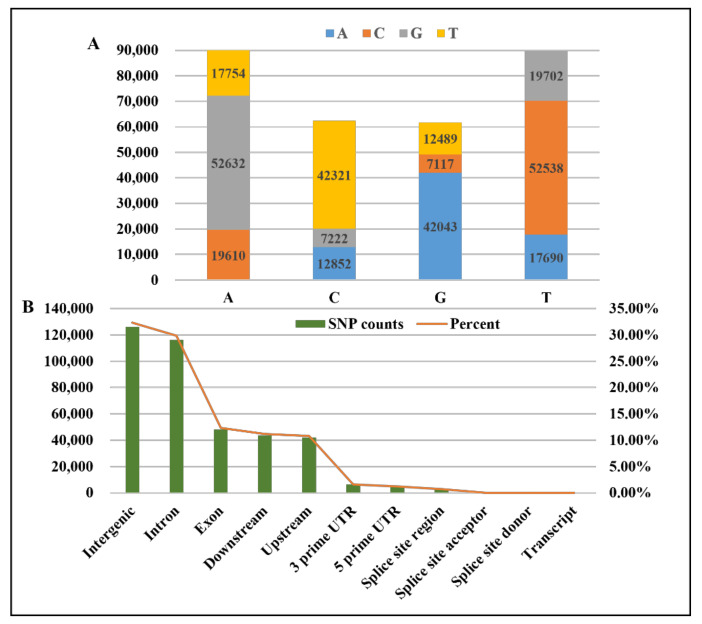
The DNA base mutation and location in the genome of differential SNPs. (**A**) DNA base mutation of differential SNPs; (**B**) location in the genome of differential SNPs.

**Figure 3 animals-12-02181-f003:**
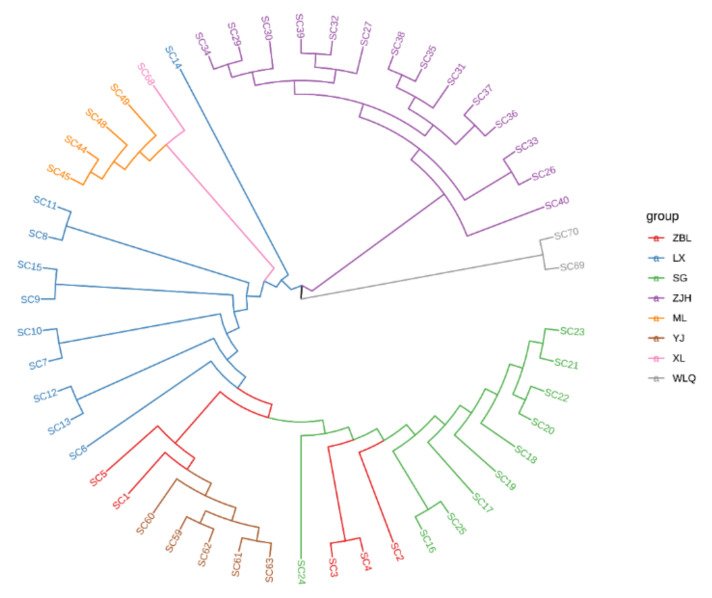
NJ phylogenetic tree of *S. kozlovi* populations. ZBL, SC1−SC5; LX, SC6−SC15; SG, SC16−SC25; ZJH, SC26−SC40; ML, SC41−SC51; YJ, SC52−SC67; XL, SC68; WLQ, SC69 and SC70. ZBL, Zhubalong; LX, Luoxu; SG, Shigu; XL, Xinlong; YJ, Yajiang; ML, Muli; ZJH, Zongjihe. The Jinsha River includes LX, ZBL, and SG. The Yalong River contains XL, YJ, and ML. The Wujiang River includes ZJH.

**Figure 4 animals-12-02181-f004:**
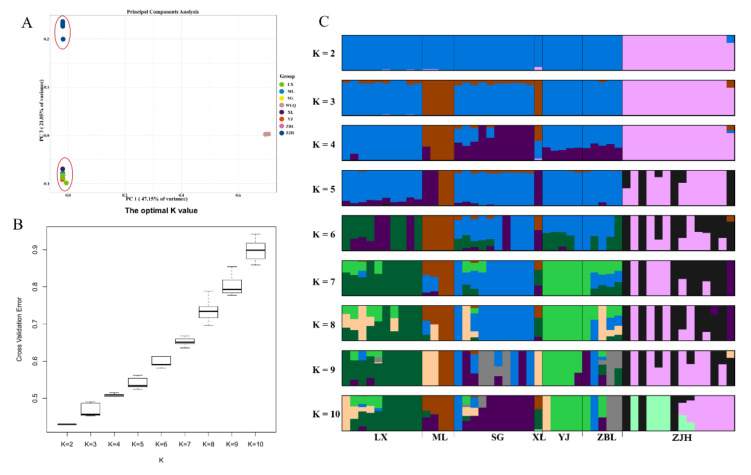
Principal component analysis and the error rate of the *S. kolzovi* admixture K value. (**A**) Principal component analysis for *S. kolzovi* based on 303,970 SNPs; (**B**) the error rate of the *S. kolzovi* admixture K value by cross−validation; (**C**) the clustering results of *S. kolzovi*. LX, Luoxu; ZBL, Zhubalong; SG, Shigu; XL, Xinlong; YJ, Yajiang; ML, Muli; ZJH, Zongjihe.

**Figure 5 animals-12-02181-f005:**
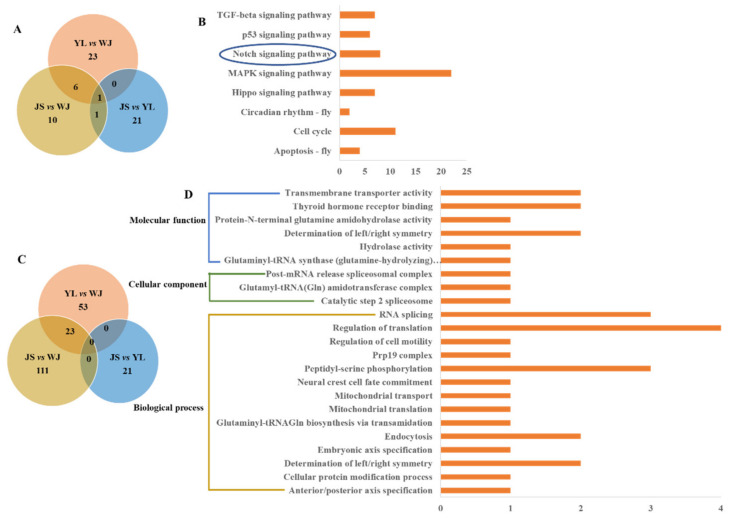
Functional annotations of the selected SNPs based on KEGG pathways and GO terms. (**A**) Venn diagram of significant KEGG pathways; (**B**) six common signaling pathways; (**C**) Venn diagram of significant GO terms; (**D**) 23 common GO terms. YL, Yalong River; JS, Jinsha River; WJ, Wujiang River. The Notch signaling pathway was common in the YL vs. WJ, JS vs. WJ, and JS vs. YL groups.

**Table 1 animals-12-02181-t001:** Specimens for GBS.

Species	Sampling Site	Abbreviation	Tributary	Coordinates	Altitude	Sample Number
*Schizothorax kozlovi*	Luoxu, Sichuan	LX	Jinsha River	E 97°59′43.47″ N 32°27′41.20″	3288	10
	Zhubalong, Sichuan	ZBL		N 29°46′17.46″ E 99°0′38.54″	2476	5
	Shigu, Sichuan	SG		E 99°57′44.69″ N 26°52′12.82″	1818	10
	Xinlong, Sichuan	XL	Yalong River	E 100°18′48.15″ N 30°56′17.73″	3052	1
	Yajiang, Sichuan	YJ		E 101° 0′58.49″ N 30° 2′0.01″	2572	5
	Muli, Sichuan	ML		E 101°15′59.47″ N 27°51′32.50″	1780	4
	Zongjihe, Guizhou	ZJH	Wujiang River	E 105°12′52.84″ N 27° 2′52.95″	1214	14
*Schizopygopsis malacanthus*	Yajiang, Sichuan	WLQ	Yalong River	E 101° 0′58.49″ N 30° 2′0.01″	2572	1
*Gymnodiptychus pachycheilus*	Xinlong, Sichuan	WLQ	Yalong River	E 100°18′48.15″ N 30°56′17.73″	3052	1

Note: WLQ represents outgroups.

**Table 2 animals-12-02181-t002:** Genetic diversity parameters among *S. kozlovi* populations.

	Populations	*H* _E_	*H* _O_	π	*PIC*
Jinsha River	LX	0.09693	0.07411	0.07827	0.06128
	ZBL	0.09175	0.06706	0.07523	0.05439
	SG	0.09312	0.07058	0.07452	0.05801
	Average value	0.09393	0.07058	0.07601	0.05789
Yalong River	XL	0.09802	0.04901	0.09202	0.03676
	YJ	0.09437	0.06695	0.07486	0.05411
	ML	0.09576	0.06614	0.07676	0.05299
	Average value	0.09605	0.06070	0.08121	0.04795
Wujiang River	ZJH	0.10054	0.07821	0.08132	0.06465

Note: *H*_E_, expected heterozygosity; *H*_O_, observed heterozygosity; *PIC*, polymorphism information content; π, nucleotide diversity; LX, Luoxu; ZBL, Zhubalong; SG, Shigu; XL, Xinlong; YJ, Yajiang; ML, Muli; ZJH, Zongjihe.

**Table 3 animals-12-02181-t003:** Genetic distance within and between *S. kozlovi* populations (above diagonal) and genetic fixation index (*F*_st_) (below diagonal) of populations.

	Pop	ZBL	LX	SG		ZJH	ML	YJ	XL	WLQ
Jiansha River	ZBL	−	0.0038	0.0053		0.2014	0.0588	0.0053	0.0512	1.5535
	LX	0.0038	−	0.0110		0.1974	0.0543	0.0074	0.0315	1.6972
	SG	0.0053	0.0109	−		0.2073	0.0549	0.0108	0.0533	1.7441
Wujiang River	ZJH	0.1824	0.1791	0.1872		−	0.2095	0.2094	0.1720	1.7522
Yalong River	ML	0.0571	0.0529	0.0534		0.1890	−		0.0606	1.4727
	YJ	0.0053	0.0073	0.0107		0.1889	0.0636	−	0.0718	1.5669
	XL	0.0499	0.0310	0.0519		0.1580	0.0588	0.0693	−	0.7783
Outgroup	WLQ	0.7885	0.8168	0.8252		0.8266	0.7707	0.7913	0.5408	−

Note: LX, Luoxu; ZBL, Zhubalong; SG, Shigu; XL, Xinlong; YJ, Yajiang; ML, Muli; ZJH, Zongjihe.

## Data Availability

Data will be available upon request.

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
