# Peer review of "Genetic Variation in Schizothorax kozlovi Nikolsky in the Upper Reaches of the Chinese Yangtze River Based on Genotyping for Simplified Genome Sequencing"

_animals, 2022, doi:10.3390/ani12172181_

Round 1

Reviewer 1 Report

It is a good study to evaluate the genetic diversity of Schizothorax kozlovi and propose actions for the conservation of this species, congratulations!

I have a cuestión, it could justify why it used the species of Schizopygopsis malacanthus and Gymnodiptychus pachycheilus as an external group, existing the possibility of using other fishes?.

Author Response

Thank you very much for your professional review of our article. Based on your suggestion, we have made the following explanations for the use of Schizopygopsis malacanthus and Gymnodiptychus pachycheilus as an external group:

First, these two species of fish (Schizopygopsis malacanthus and Gymnodiptychus pachycheilus) and the experimental object of this paper, Schizothorax kozlovi, belong to Schizothoracinae and are closely related, which can better verify the genetic differentiation of Sichuan schizostomy.

Second, Schizopygopsis malacanthus and Gymnodiptychus pachycheilus were easily collected in the Jinsha River and Yalong River, and all GBS sequencing samples need to be sequenced on the same platform in the same time period to ensure more accurate sequencing results.

Finally, in studies on the genetic evolution of Schizothoracinae fish, many studies have used fish in the same subfamily as alien groups for genetic differentiation verification, such as Schizothorax curilabiatus, which was used as an outlier group for genetic diversity analysis of Schizothorax molesworthi from the lower reaches of the Yarlung Zangbo River and the Chayu River (Yu et al., 2019). In a comparative study on the genetic differences and differentiation levels of six Diptychus maculates in the Yili River and Tarim River systems, Gymnocypris eckloni and Schizopygopsis pylzovi were used as outgroups (Yang et al., 2014).

As described in the review, it is no question of using Schizothoracinae fish as outgroups in this study, provided that they are collected and sequenced at the same time as the experimental subjects in this study.

Reviewer 2 Report

The authors analyzed different populations of Schizothorax kozlovi by GBS, which contributes to a better understanding of their evolutionary relationships.

Overall, the manuscript is well-written and provides well-organized information.

I'd suggest Authors the following modifications:

-shorten the abstract: it is too long

- improve the introduction, by providing some description/information more about GBS (not all readers are so familiar with such a technique)

-some comparison across other -also not related- species (if there are available data in the literature) of the loci/variants found to be variable across environments with different temperatures might be very insightful and improve the manuscript

Author Response

On behalf of all the contributing authors, I would like to express our sincere appreciation for your letter and the reviewers’ constructive comments concerning our article entitled “Genetic Variation in Schizothorax kozlovi Nikolsky in the Upper Reaches of the Chinese Yangtze River Based on Geno-typing for Simplified Genome Sequencing” (Manuscript No: animals-1825170). These comments are valuable and helpful for improving our manuscript. In accordance with your comments, we substantially revised the manuscript, supplemented the introduction of GBS sequencing, and discussed the genetic differentiation of other species at different temperatures, making our results convincing. The results of this study will increase our understanding of the diversity of S. kozlovi in the upper reaches of the Yangtze River and provide a basis for the conservation and utilization of wild resources. In this revised version, our changes to the manuscript are highlighted in red text in the document. A point-by-point response to this excellent reviewer is listed below the letter.

Point 1: Shorten the abstract: it is too long

Response 1: L81 We thank the reviewer for the suggestion, and we have made the sentences of the abstract more brief and scientific. The revised abstract is as follows:

Abstract: Schizothorax kozlovi Nikolsky is a unique cold-water fish in the upper reaches of the Yangtze River in China that has high economic value. To date, there has been no systematic study on the genetic diversity and population structure of S. kozlovi using genotyping by sequencing (GBS). In our study, genetic diversity and population structure analyses were performed on seven wild populations (originating from the Jinsha River, Yalong River, and Wujiang River) in the upper reaches of the Yangtze River by GBS. The results indicated that a total of 303,970 single-nucleotide polymorphisms (SNPs) were identified from the seven wild populations. Lower genetic diversity was exhibited among the intrapopulations of the three tributaries. The neighbor-joining (NJ) phylogenetic tree was consistent with the principal coordinate analysis (PCA) results, supporting the structure analysis, which indicated that the Wujiang River population had significant genetic differentiation when compared to the Jinsha River and Yalong River populations. Furthermore, the selected SNPs were enriched in cellular processes, environmental adaptation, signal transduction, and related metabolic processes between the Wujiang population and the other two populations. The above results indicate that the populations of S. kozlovi have different degrees of tolerance and selection pressure in response to temperature and altitude. The Wujiang population was genetically differentiated from the Jinsha River and Yalong River populations. The Wujiang intrapopulation has greater genetic diversity and differentiation than the Jinsha River and Yalong River populations, which demonstrates that the Jinsha and Yalong populations require more attention and resources for their protection. The results of this study will increase our understanding of the diversity of S. kozlovi in the upper reaches of the Yangtze River and provide a basis for the conservation and utilization of wild resources.

Point 2: Improve the introduction, by providing some description/information more about GBS (not all readers are so familiar with such a technique)

Response 2: L101 —109 Thank you for the reviewer's advice on writing the introduction. GBS technology has not yet been popularized in Schizothoracinae fish. Therefore, we have added a detailed description of the GBS technology to the article:

Genotyping by sequencing (GBS), as one of the simplified genome sequencing approaches, can identify many single-nucleotide polymorphisms (SNPs) to reveal the evolutionary relationship of species and population genetics[10-12]. Unlike single-molecule DNA sequencing methods, a highly promising GBS approach can discover tens of thousands of variants and obtain genotypic information at the genomic level [13]. GBS reduces the complexity of genome sequencing through restriction endonucleases that are particularly suitable for species with large genomes, high repetition frequencies, and limited genomic resources[13]. Furthermore, SNPs from GBS are much more abundant, have lower mutation rates, and can be genotyped with lower error rates[14]. In previous studies, GBS has been used to solve a series of evolutionary biological problems, such as fish genetic protection and local adaptation of different populations[15, 16].

Point 3: Some comparison across other -also not related- species (if there are available data in the literature) of the loci/variants found to be variable across environments with different temperatures might be very insightful and improve the manuscript.

Response 3: L342—346 Our thanks to the reviewer for the suggestion to add to the discussion. In the discussion section of this paper, we have added examples of temperature adaptation in fish of the subfamily Schizothoracinae fish (Diptychus maculates) as follows:

The temperature and altitude of different habitats reflect the differences in their adaptation to the environment[48]. Previous studies have shown that the genetic isolation of Diptychus maculates is significantly related to temperature, and the large variation in environmental factors leads to weak gene flow in the species[48].

Round 2

Reviewer 2 Report

While appreciating the authors' efforts in improving the manuscript, I'd like to point out that:

1) the abstract might be further shortened, omitting many details that summarize/anticipate the results (it is not mandatory, only a suggestion).

2) my suggestion was about the comparison between the variable genetic traits (e.g. SNP) found to be associated with the temperature in S.k. and those -if any- reported in the literature as associated with the temperature in other species, to discuss this point (it would be very interesting). The Authors probably have misunderstood my request.

Author Response

Thanks again to the reviewers for their professional comments. In this revised version, our changes to the manuscript are highlighted in red text in the document. A point-by-point response to this excellent reviewer is listed below the letter.

Point 1: The abstract might be further shortened, omitting many details that summarize/anticipate the results (it is not mandatory, only a suggestion).

Response 1: L28- We thank the reviewer for the suggestion, and we have made the sentences of the abstract more brief and scientific. The revised abstract is as follows:

Abstract: Schizothorax kozlovi Nikolsky is a unique cold-water fish in the upper reaches of the Yangtze River in China and has high economic value. In our study, genetic diversity and population structure analyses were performed on seven wild populations (originating from the Jinsha River, Yalong River, and Wujiang River) in the upper reaches of the Yangtze River by genotyping by sequencing (GBS). The results indicated that a total of 303,970 single-nucleotide polymorphisms (SNPs) were identified from the seven wild populations. Lower genetic diversity was exhibited among the intrapopulations of the three tributaries, and the Wujiang River population had significant genetic differentiation when compared to the Jinsha River and Yalong River populations. Furthermore, the selected SNPs were enriched in cellular processes, environmental adaptation, signal transduction, and related metabolic processes between the Wujiang population and the other two populations. The above results indicate that the populations of S. kozlovi have different degrees of tolerance and selection pressure in response to temperature and altitude. The Wujiang intra-population has greater genetic diversity and differentiation than the Jinsha River and Yalong River populations, which demonstrates that the Jinsha and Yalong populations require more attention and resources for their protection. The results of this study will increase our understanding of the diversity of S. kozlovi in the upper reaches of the Yangtze River and provide a basis for the con-servation and utilization of wild resources..

Point 2: My suggestion was about the comparison between the variable genetic traits (e.g. SNP) found to be associated with the temperature in S.k and those -if any- reported in the literature as associated with the temperature in other species, to discuss this point (it would be very interesting). The Authors probably have misunderstood my request.

Response 2: L415 —426 We thank the reviewer for the suggestion. In the discussion section of this paper, we have added examples of the comparison between the variable genetic traits in temperature as follows:

The temperature-related SNPS found in this study are mainly involved in the functions of genes, including mitochondrial translation and transport, energy metabolism, etc. It is possible that temperature is a stimulus to S. kozlovi, causing various physiological reactions, and thus gene transcription become very active. High mitochondrial gene concentration has been studied in other cold-adapted ectotherms. Windisch demonstrated that the mitochondrial ribosomal protein in Pachycara brachycephalum has a high transcription rate under cold conditions, which has a positive effect on the adaptation of antarctic fish to cold environment [54]. In addition, increased energy requirements under temperature stress are usually required to maintain normal body function, thus allocating less energy to growth, storage, and reproduction [55]. In aquatic ectotherms such as mussels, extreme high temperatures will lead to a reduction in aerobic range or induce a reduced metabolic rate [56].